Women are underrepresented on the editorial boards of journals in environmental biology and natural resource management

Cho Alyssa H. 1
Johnson Shelly A. 2
Schuman Carrie E. 3
Adler Jennifer M. 3
Gonzalez Oscar 3
Graves Sarah J. 2
Huebner Jana R. 3
Marchant D. Blaine 4
Rifai Sami W. 2
Skinner Irina 5
Bruna Emilio M. 5 6 embruna@ufl.edu
1 Agronomy Department, University of Florida , Gainesville, FL , USA
2 School of Forest Resources and Conservation, University of Florida , Gainesville, FL , USA
3 School of Natural Resources and Environment, University of Florida , Gainesville, FL , USA
4 Biology Department, University of Florida , Gainesville, FL , USA
5 Department of Wildlife Ecology & Conservation, University of Florida , Gainesville, FL , USA
6 Center for Latin American Studies, University of Florida , Gainesville, FL , USA
Lortie Christopher
Electronic publication date: 2014 Aug 21
Publication date: 2014
Volume: 2
Electronic Location ID: e542
Received 2014 May 29; Accepted 2014 Aug 5
Copyright: © 2014 Cho et al.
Copyright year: 2014
Copyright holder: Cho et al.
License: This is an open access article distributed under the terms of the Creative Commons Attribution License, which permits unrestricted use, distribution, reproduction and adaptation in any medium and for any purpose provided that it is properly attributed. For attribution, the original author(s), title, publication source (PeerJ) and either DOI or URL of the article must be cited.
License URL: https://creativecommons.org/licenses/by/4.0/

Keywords: Gender, Editorial Boards, Bias, Associate Editors, Subject Editors, Editor-in-Chief

Funding: UF Center for Latin American Studies Faculty Research Award Dean of the University of Florida’s College of Agricultural and Life Sciences Interim Deans for Research and Directors of the Florida Agricultural Experiment Station Support for data collection was provided by a UF Center for Latin American Studies Faculty Research Award to Emilio Bruna. Support for publishing this manuscript in an Open Access journal was provided by the Dean of the University of Florida’s College of Agricultural and Life Sciences and the Interim Deans for Research and Directors of the Florida Agricultural Experiment Station. The funders had no role in study design, data collection and analysis, decision to publish, or preparation of the manuscript.

==============================
Despite women earning similar numbers of graduate degrees as men in STEM disciplines, they are underrepresented in upper level positions in both academia and industry. Editorial board memberships are an important example of such positions; membership is both a professional honor in recognition of achievement and an opportunity for professional advancement. We surveyed 10 highly regarded journals in environmental biology, natural resource management, and plant sciences to quantify the number of women on their editorial boards and in positions of editorial leadership (i.e., Associate Editors and Editors-in-Chief) from 1985 to 2013. We found that during this time period only 16% of subject editors were women, with more pronounced disparities in positions of editorial leadership. Although the trend was towards improvement over time, there was surprising variation between journals, including those with similar disciplinary foci. While demographic changes in academia may reduce these disparities over time, we argue journals should proactively strive for gender parity on their editorial boards. This will both increase the number of women afforded the opportunities and benefits that accompany board membership and increase the number of role models and potential mentors for early-career scientists and students.

Introduction

Despite women in the United States and Europe earning similar numbers of graduate degrees as men do, they remain underrepresented in upper level positions in both academia and industry in these regions (European Commission, 2012; National Science Foundation, 2004; National Science Foundation, 2012). Several mechanisms have been put forward to explain this disparity, including biases against women in hiring, promotion, and offers of compensation, the emphasis on productivity, journal placement, and citation rates as determinants of merit despite evidence of gender bias influencing all three, inflexible or even hostile work environments, and a lack of role models and mentors (reviewed in Budden et al., 2008; Lariviere et al., 2013; Leahey, 2007; Long, 2001; Moss-Racusin et al., 2012). In response, universities, funding agencies, and other institutions have implemented strategies to address these issues, including making opportunities for professional advancement more broadly available and actively seeking gender diversity in leadership roles (Fox, 2008). While these efforts have had some positive results, much remains to be done to ensure women in Science, Technology, Engineering, and Math (STEM) disciplines are afforded the same opportunities as their male counterparts.

The editorial boards of scientific journals act as gatekeepers that help maintain the scientific integrity and standards of a journal as well as identify emerging and innovative areas of research (Addis & Villa, 2003; Mauleon et al., 2013). An invitation to serve as a Subject Editor is recognition that a scholar is respected in his or her discipline; it is also the path towards leadership positions because Associate Editors and Editors-in-Chief are typically selected from the Subject Editors. Serving on a board is also a means of advancing one’s scholarship, both by becoming aware of the latest advances in the field and gaining insights into the writing and publication process. Finally, editorial boards are important professional networks—in serving on a board one is able to develop relationships with reviewers, authors, and other editors (Addis & Villa, 2003; Pearson et al., 2006). Serving on a board is therefore both an honor and a means of furthering one’s research and career.

Previous studies have quantified the gender composition of editorial boards in the social sciences (Addis & Villa, 2003; Green, 1998; Stark et al., 1997), business administration and management (Metz & Harzing, 2012), and STEM fields such as information systems (Cabanac, 2012) and medicine (Galley & Colvin, 2013; Keiser, Utzinger & Singer, 2003; Wilkes & Kravitz, 1995). To our knowledge, however, no such efforts have been made in ecology, natural resource management, plant sciences, or related disciplines (collectively referred to here as “environmental biology”). We therefore used ten highly regarded journals in environmental biology to address the following questions: (1) What proportion of editorial board members were women between 1985 and 2013? (2) How did the representation of women on editorial boards change over this time period? (3) How many women served in leadership positions, i.e., as Editors-in-Chief or Associate Editors?

Methods

We selected for review 10 high profile environmental biology journals: Annual Review of Ecology, Evolution, and Systematics, Biotropica, Agronomy Journal, North American Journal of Fisheries Management, American Journal of Botany, Conservation Biology, Biological Conservation, Ecology, Journal of Ecology, and Journal of Tropical Ecology. We chose these journals because they are published by the primary professional organizations of which we (i.e., the authors) are members (e.g., Biotropica, Conservation Biology) or are alternative, non-society outlets for similar research (e.g., Journal of Tropical Ecology, Biological Conservation). It was not intended to be a random sample of journals or a subset of journals with similar impact factors. Rather, they were chosen because they are the journals that many graduate students in environmental biology, natural resource management, and plant sciences, including the authors, target to publish some of their thesis research.

Our analyses were based on the years 1985–2013. We chose 1985 as a starting point because it is shortly after studies began demonstrating disparities in career advancement between male and female scientists (reviewed in Long, 2001; National Science Foundation, 2003) but a few years prior to major initiatives by the US National Science Foundation and others to rectify these disparities (e.g., the 2001 initiation of the ADVANCE Program, National Science Foundation, 2014). As such, we expect our survey period to reflect potential shifts in editorial board composition resulting from increased awareness of gender biases in STEM and the results of efforts to rectify these biases. For each journal we selected the first issue published each year and recorded the names, institutions, and editorial positions of all editorial board members. We then used Internet searches, personal knowledge, and interviews of colleagues to determine the gender of each editorial board member. Because of library licensing issues we were unable to obtain data for Journal of Tropical Ecology for the years 1986–1989.

Journals often have different names for positions with similar editorial responsibilities, these names frequently change over time, and editorial positions are frequently created or eliminated. We therefore assigned editorial board members to the following categories based on their responsibilities: (1) Editor-in-Chief (EIC). The EIC oversees the journal and is ultimately responsible for editorial policy, standards, and practice, including appointing members of the Editorial Board. Some journals (e.g., North American Journal of Fisheries Management) had co-Editors-in-Chief; in such cases all were included in the total EIC count. (2) Associate Editors (AE). The AE assists the EIC with their responsibilities and may take the lead on some aspects of journal administration; in some cases they might oversee all submissions in a particular subject area or from a geographic region. Not all journals have AEs, while those that do may vary in the length of time they have had them. (3) Subject Editors (SE). The SEs, also commonly referred to as Handling Editors, oversee the process of manuscript review. For some journals they make final decisions on manuscripts after considering reviewer feedback (e.g., Ecology) while for others they provide recommendations based on which EICs or AEs make final decisions (e.g., Biotropica). They also provide formal or informal feedback to the EICs/AEs on journal policy and administration. They are referred to collectively by a variety of titles, including Board of Editors (Ecology, Biological Conservation) and the Editorial Committee (Annual Review of Ecology, Evolution, and Systematic, American Journal of Botany). Note that two journals—the American Journal of Botany and North American Journal of Fisheries Management—used the title of “Associate Editor” for members of their Editorial Board with the responsibilities of SEs; we therefore included them in this category in our analyses. (4) Special Editors. Many journals have someone tasked with organizing special sections, reviewing data archives, soliciting reviews of recently published books of interest to the journal’s readers, etc. (e.g., Biological Florida Editors for the Journal of Ecology; Concept Section, Data Archive, Special Features, and Invited Papers Editors for Ecology).

We conducted our analyses using only EICs, AEs, and SEs, and throughout our manuscript and analyses we use the term ‘Editorial Board’ to refer to the group collectively made up of these three categories. Special Editors were not included in our analyses unless they were also identified as EICs, AEs, or SEs because very few journals had these positions and they rarely existed for the entire survey period. We also excluded from our analyses production staff (e.g., production editors, managing editors, editorial assistants) and the American Journal of Botany’s “Section Representatives”, which were only present in our survey in 1985 and whose primary function was to help identify journal priorities and suggest reviewers if asked—they did not make editorial decisions on individual manuscripts (JE Skog, pers. comm., 2014). Analyses were conducted with R version 3.1.0 (R Core Development Team, 2014).

Results & Discussion

We found that from 1985–2013 only 16% of subject editors were women (N = 332 of 2065, Fig. 1A). The disparity also extends to leadership positions: since 1985 only 14% of Associate Editors (N = 18 of 125, Fig. 1B) and 12% of the Editors-in-Chief of our focal journals were women (N = 7 of 59, Fig. 1C). Not surprisingly, the proportions of male and female editors were significantly different for all of groups of Editors (proportion tests with continuity corrections, null probability = 0.5, SE: χ2 = 946.44, df = 1, p < 0.0001; AE: χ2 = 61.952, df = 1, p < 0.0001; EIC: χ2 = 32.81, df = 1, p < 0.0001).

Figure 1 Gender representation on 10 editorial boards in environmental biology.

The proportion of men and women who served as (A) Subject Editors, (B) Associate Editors, and (C) Editors-in-Chief of 10 environmental biology journals from 1985 to 2013.

While there was a general increase in the representation of women on editorial boards over time, for most journals the percentage of women on the board rarely exceeded 20% (Fig. 2). Nevertheless, there was notable variation among journals in the representation of gender on their editorial boards during the time period surveyed. For several journals, the proportion of women editors increased from zero in the mid-1980s to ∼40% by 2013 (e.g., Biotropica, American Journal of Botany, Conservation Biology). Others, however, had consistently few women on their boards throughout the period surveyed (e.g., Agronomy Journal, North American Journal of Fisheries Management, Biological Conservation). A similar pattern of underrepresentation was observed for Associate Editors and Editors-in-Chief. While most journals had female Associate Editors at some point during the period surveyed, only 5 of the 10 journals we reviewed had ever had a woman as Editor-in-Chief (Fig. 3). Of these, only one—the North American Journal of Fisheries Management—had multiple women serve as EICs.

Figure 2 Change in the percentage of women on 10 editorial boards from 1985 to 2013.

Editorial boards comprise Editors-in-Chief, Associate Editors, and Subject Editors. The dashed line represents a 50:50 gender ratio.

Figure 3 Total number of men and women who served as (A) Editors-in-Chief (B) Associate Editors or (C) Subject Editors of 10 environmental biology journals.

Data are from 1985 to 2013. Note that we categorized the Associate Editors of the American Journal of Botany and North American Journal of Fisheries Management as Subject Editors given their responsibilities, and hence they are depicted with that category (B).

We recognize that determining the extent of gender bias in the composition of editorial boards in environmental biology will require evaluating many more journals from multiple subfields. However, the results of similar surveys in fields ranging from economics to anthropology have found disparities comparable to those we document (Addis & Villa, 2003; Cabanac, 2012; Galley & Colvin, 2013; Green, 1998; Keiser, Utzinger & Singer, 2003; Metz & Harzing, 2012). Assuming the results for the journals we reviewed are representative of others in environmental biology, our observations suggest two questions to be addressed by this scientific community. First, why are women underrepresented on editorial boards and in positions of editorial leadership? Second, for what gender composition on editorial boards should journals strive?

While our study was not designed to elucidate why women are underrepresented on editorial boards, potential mechanisms include many of the same ones that are invoked to explain why women are lacking in leadership positions in other spheres of academia (Fox, 2008; Long, 2001). It may also be that men continue to be more visible and hence more likely to be identified as potential board members because they have greater productivity, have more first- or last-authors of papers (West et al., 2013), and tend to be “citation elites” (sensu Parker, Allesina & Lortie, 2013; Parker, Lortie & Allesina, 2010). It may be that using these metrics to screen for editors might eventually—albeit slowly—result in increased numbers of women on editorial boards. This is because gender-based disparities in rates of publication (West et al., 2013) and citation (Borsuk et al., 2009) are diminishing (but see Lariviere et al., 2013), although this does not appear to be the case for all disciplines (West et al., 2013). More difficult to overcome might be the reliance on using the social and research networks of (mostly male) editorial board members to identify potential new editors (Addis & Villa, 2003), since women scientists are frequently excluded from such networks or on their periphery (Fox, 2008). This is where proactive measures, including the promotion of women to positions of editorial leadership, may have the greatest impact (Galley & Colvin, 2013). Indeed, at least one study has found that having a female Editor-in-Chief is correlated with a greater proportion of women on editorial boards (Mauleon et al., 2013).

For what gender composition on editorial boards should journals in environmental biology strive? We propose they should proactively seek gender parity, rather than simply mirror the proportion of women earning doctoral degrees in a specialization, conducting research in particular disciplines, or who are members of academic societies—numbers which, in contrast to other fields (e.g., Morton & Sonnad, 2007), we were surprised to find are extremely difficult to ascertain for environmental biology. Some might argue that the relatively lower number of female senior scholars in certain fields (e.g., agronomy) might make parity a challenge. However, it is important to emphasize that the issue is not whether there is parity in the number of women earning PhDs, but whether there are sufficient qualified women worldwide to comprise half an editorial board, which is a much smaller number (mean number of board members in 2012 = 56 ± 41.3 SD, range = 9–127). It is difficult to argue that there are not, given the global reach of academic societies (Carroll, 2014), the internationalization of research programs (Stocks et al., 2008), increases in research productivity in developing countries (Holmgren & Schnitzer, 2004), and the time elapsed since issues of gender & STEM came to the fore (though we concede that for highly specialized or national journals parity may be a greater challenge). We argue that Editors must work harder to proactively identify these potential board members—the fact that journals with similar disciplinary foci can have very different representation (e.g., Biological Conservation and Conservation Biology, Biotropica and Journal of Tropical Ecology) suggests increasing the proportion of women on editorial boards can be matter of policy and not pool size.

Attempts by journals to strive for gender parity would greatly increase the number of women afforded the opportunities and benefits that accompany board membership, as well as increase the number of female role models and mentors for early-career scientists and students seeking guidance on scientific publishing. When coupled with initiatives such as double-blind reviewing (Budden et al., 2008) and efforts to explore factors that influence our perceptions of ‘merit’ (Lortie et al., 2007), editorial board parity could ultimately help reduce the pervasive and insidious “gender productivity puzzle” first identified over thirty years ago (Cole & Zuckerman, 1984). Finally, a more inclusive editorial board might bring unanticipated benefits to the journal itself, including attracting a broader diversity of research topics, contributors, and approaches (Stegmaier, Palmer & van Assendelft, 2011). All of this could greatly increase a journal’s impact via shaping both the discipline and the scientific workforce advancing it.

We thank R Primack, R Eades, C Lortie, J Parker, and F Piper for helpful discussions or feedback on the manuscript and M Duryea, D Archer (Interim Deans for Research and Directors of the Florida Agricultural Experiment Station), and RE Turner (Dean of the College of Agricultural and Life Sciences) for funds to publish in an open-access journal.

Additional Information and Declarations

Competing Interests

Author Contributions

Data Deposition

Emilio M. Bruna is Editor-in-Chief of Biotropica, which is one of the journals surveyed for this manuscript.

Alyssa H. Cho, Shelly A. Johnson, Carrie E. Schuman, Jennifer M. Adler, Oscar Gonzalez, Sarah J. Graves, Jana R. Huebner, D. Blaine Marchant, Sami W. Rifai and Irina Skinner conceived and designed the experiments, performed the experiments, analyzed the data, wrote the paper, reviewed drafts of the paper.

Emilio M. Bruna conceived and designed the experiments, performed the experiments, analyzed the data, contributed reagents/materials/analysis tools, wrote the paper, prepared figures and/or tables, reviewed drafts of the paper.

The following information was supplied regarding the deposition of related data and computer code.

Dryad Digital Repository, http://doi:10.5061/dryad.6jn86: Alyssa H. Cho, Shelly A. Johnson, Carrie E. Schuman, Jennifer M. Adler, Oscar Gonzalez, Sarah J. Graves, Jana R. Huebner, D. Blaine Marchant, Sami W. Rifai, Irina Skinner and Emilio M. Bruna 2014. Women are underrepresented on the editorial boards of journals in environmental biology and natural resource management.

R code for analyses and figures in this paper are available at GitHub.com: https://github.com/embruna/Editorial-Board-Gender.

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
