# Peer review of "Women are underrepresented on the editorial boards of journals in environmental biology and natural resource management"

_PeerJ, doi:10.7717/peerj.542_

## Round 0.1 · original submission · Minor Revisions

I think this paper is an important contribution. However, please see the comments by the referees. I think they are very useful, and I would like to see more development of the implications/explanations as well.

cheers,
Chris.

·

Basic reporting

This article recapitulates findings from other areas and disciplines with respect to the general composition differential of editorial boards - an important and worthy topic. Still, I would like to see several additions. First, it would be good to speculate just a bit about why this is an issue within the environmental sciences specifically. The authors mention a few possibilities in the intro but it would be nice to connect more concretely to what we know about gender dynamics in these fields - for instance, we know that the vast majority of citation elites in these fields are male (Parker et al. 2010, 2013), mostly because it takes time for citations to accrue and science was in the not so distant past dominated by men. So what can we expect in the future? Will these compositions become more equal as women matriculate to higher positions and become more influential? Also, how do these numbers relate to the proportion of senior women in these fields generally?

Second, if possible, you might also offer a few thoughts about why the subject categories represented by these journals differ so markedly - what is driving these subfield trends? Also, if you arrange the journals by level of prestige (IF, flagship ESA, whatever) are there trends apparent across that continuum? Are there few women in these positions as we move up the journal status hierarchy?

Third, it would be good to include precise percentages on your pie charts so the reader knowns exact numbers with a glance at the figure and does not have to go into the text.

Finally, read and integrate some cognate work on this topic. Here I am thinking of work by Amber Budden on editorial boards and peer review decisions, work by J. Scott Long on women and social stratification in science, and work by Erin Leahey on women in the sciences.

Experimental design

No comments

Validity of the findings

No Comments

·

Basic reporting

The manuscript is well written and exposes clearly the importance of the subject. There are few sentences where I think the English grammar may be not correct, although English is not my original language and I may be wrong. I have highlighted these parts in the text.
The introduction would benefit for more details about previous work. After reading it, one is not clear whether this is the first study addressing gender representation in Editorial boards of STEM disciplines. Either if it is or not the first one, it should be stated why is it necessary to know this. I know that the implicit idea of the authors may be that "to correct something we need first to know how bad it is", but that message must be clearly established.
The criteria for journals selection is not clear to me.
Results are clear and well presented, I liked the figures. However, I think that statistics is necessary in this study. I recommend a proportion test at least. I also wonder whether some analysis discriminating journals by impact factors or by sub discipline could be addressed. It seems interesting to know if women are similarly poorly represented in high-impact journals as in low impact ones. It would be also interesting to consider the number of undergraduate students across sub-disciplines. In agronomy, for example, the aim of gender equality can face the problem of a limited number of women available to do editorial activities. That type of limitations is less probable in disciplines like biology or medicine.

Experimental design

I recommend a better justification of the criteria for journal selection. I did not understand this point, as I mention in the document.

Validity of the findings

Although the differences between gender illustrated by the figures (proportions) are robust, statistics must be incorporated. There is no statistic at all in the study.

Additional comments

Attached the authors can find many minor and also major comments on the manuscript. I think this is an interesting study, but needs to be better contextualized. I hope my comments help to achieve a strong context.

---

## Round 0.2 · accepted · Accept

Thanks. I would have still liked to see more of the speculation and linkages to social sciences as proposed by Dr Parker, but the manuscript is greatly improved. Thanks.